# Hybrid Coils-Based Wireless Power Transfer for Intelligent Sensors

**DOI:** 10.3390/s20092549

**Published:** 2020-04-30

**Authors:** Mustafa F. Mahmood, Saleem Lateef Mohammed, Sadik Kamel Gharghan, Ali Al-Naji, Javaan Chahl

**Affiliations:** 1Department of Medical Instrumentation Techniques Engineering, Electrical Engineering Technical College, Middle Technical University, Baghdad 10001, Iraq; musiraq86@gmail.com (M.F.M.); saleem_lateef_mohammed@mtu.edu.iq (S.L.M.); sadik.gharghan@mtu.edu.iq (S.K.G.); 2School of Engineering, University of South Australia, Mawson Lakes, SA 5095, Australia; Javaan.Chahl@unisa.edu.au; 3Joint and Operations Analysis Division, Defence Science and Technology Group, Melbourne, VIC 3207, Australia

**Keywords:** arduino, heart rate sensor, nRF24L01, transfer efficiency, transfer power

## Abstract

Most wearable intelligent biomedical sensors are battery-powered. The batteries are large and relatively heavy, adding to the volume of wearable sensors, especially when implanted. In addition, the batteries have limited capacity, requiring periodic charging, as well as a limited life, requiring potentially invasive replacement. This paper aims to design and implement a prototype energy harvesting technique based on wireless power transfer/magnetic resonator coupling (WPT/MRC) to overcome the battery power problem by supplying adequate power for a heart rate sensor. We optimized transfer power and efficiency at different distances between transmitter and receiver coils. The proposed MRC consists of three units: power, measurement, and monitoring. The power unit included transmitter and receiver coils. The measurement unit consisted of an Arduino Nano microcontroller, a heart rate sensor, and used the nRF24L01 wireless protocol. The experimental monitoring unit was supported by a laptop to monitor the heart rate measurement in real-time. Three coil topologies: spiral–spiral, spider–spider, and spiral–spider were implemented for testing. These topologies were examined to explore which would be the best for the application by providing the highest transfer power and efficiency. The spiral–spider topology achieved the highest transfer power and efficiency with 10 W at 87%, respectively over a 5 cm air gap between transmitter and receiver coils when a 200 Ω resistive load was considered. Whereas, the spider–spider topology accomplished 7 W and 93% transfer power and efficiency at the same airgap and resistive load. The proposed topologies were superior to previous studies in terms of transfer power, efficiency and distance.

## 1. Introduction

Wireless power transfer has recently found its a way into the wearable/implanted medical devices, the commercial market for electric vehicle charging facilities, mobile phones, portable laptops, smartwatches, etc. [1,2]. Medical devices are applications that use wireless power transfer technology (WPT) more than other applications, the purpose for that is removing weight and/or size from the batteries. Batteries have limited capacity, limited charge/discharge cycles, and limited efficiency with which the energy is converted. Almost any device implanted in the body requires surgery to replace the battery [2]. WPT is used in medical applications to supply power to invasive and non-invasive medical sensors or devices measuring medical vital signs. For instance, blood pressure monitors, cardiac defibrillators, electrocardiograms, electromyography, thermometers, pacemakers, neural stimulators, heart rate sensors, and glucose meters are commonly used to improve the quality of life of millions of patients [3,4]. In addition, there are several sources from which to extract energy from the host or the environment to power wearable devices such as biomass, solar, radio frequency, wind, hydro, vibration, heat energy, etc. [4]. There are two kinds of WPT, near and far field. A near field WPT acts over an air gap between transmitter and receiver sensors over short distances, <30 cm, using principles such as capacitive coupling (CC), inductive coupling (IC), magnetic resonator coupling (MRC), and the ultrasonic sensing solution (USS), which is mechanical coupling through pressure. A far field WPT acts across the air gap between transmitter and receiver sensors over long-distance, >30 cm, often using radio frequency (RF). WPT techniques used in medical applications have been the subject of animal studies to explore side effects on tissues when implanted [2,3]. Other studies have used the WPT technique at different frequencies to optimize transfer efficiency while studying side effects on skin for implantable medical devices [5]. WPT has been used in magnetic resonance imaging coils for improved magnetic field intensity on the patient side [6].

Some investigators [7,8] have used WPT based on Q-factor for improving a transfer power to operate implanted medical devices. One study, used a new circuit design that depends on capacitance to optimize transfer power and distance [9]. Other researchers used four coils to improve power transfer and efficiency for biomedical capsule endoscopy [10]. In [11], using a zero voltage switching (ZVS) component at the air gap between the transmitter and receiver coils to operate implanted medical devices. In [12], the coil was implanted inside the animal body of a pig to study the transfer power, efficiency and field effects on the animal’s tissue. Other investigators [13] used WPT to supply a transfer power for an implanted device such as pacemaker at high frequency to ignore a battery. Other studies, used WPT to supply an implantable left ventricular assist device to recharge a battery across at transfer distance between coils [14,15]. In [16], a WPT technique for an implantable medical device, such as pacemaker, was used to charge a battery based on special circuit design and different frequencies to optimize transfer efficiency and power. In [17], they used a simulated program, such as software multiphysics analysis package COMSOL, to study the effect of heat generated by coils in three configurations and transfer power to charge a battery for an implantable medical device such as pacemaker device. Used a gold coil in a WPT technique for an implantable in the brain was used to optimize transfer efficiency and power at transfer distance [18]. In [19], using two types of coil, such as a single-tube loop coil (STLC) and multi-turn copper wire coil (MTCWC), was used in a WPT for home mobile device battery charging. The contributions of this study are highlighted as follows:Designed and implemented of an MRC prototype to supply adequate power to a medical or sensor device system consisting of a wearable heart rate sensor, Arduino Nano microcontroller, and an nRF24L01 wireless protocol module.Hybrid topologies between spider and spiral coils were investigated in terms of transfer power, efficiency, and distance.Used these coils to transfer power from the transmitter coil to the receiver coil for operating the measurement unit then sends the measurement data to the monitoring unit by using a wireless protocol.Compared current study with previous studies according to the hybrid topology of coils in terms of transfer power, efficiency, and distance between transmitter and receiver.

The rest of the paper is as follows. Related work such as that for the performance metrics of the transfer energy is discussed in Section 2. Section 3 presents the proposed system design. Section 4 the designs of the coils. Section 5 describes and introduces the system experiment configurations. Section 6 highlights the overall results and discussion of the MRC system. A comparison between all topologies at several parameters is in Section 7. The results of the proposed system are compared with that of previous related works in Section 8. Finally, conclusions with trends for future work are drawn in Section 9.

## 2. Related Work

Recently, some studies have considered wireless power transfer (WPT) in medical applications. Ramachandran et al. [7] designed and implemented a wireless power transfer (WPT) system using MRC to light a neon lamp. The system consists of a power source, transmitting antenna, receiving antenna, and neon light. Experimental results indicated that resonant coupling through Q-factor analysis is superior to non-resonant coupling. A small air gap and high turn ratio leads to stronger coupling and a higher Q- factor. The system achieved a transfer distance in the range of 1–15 cm using low cost hardware. Advantages included low cost and complexity. Disadvantages were coupling losses, a short distance, and a large size. RamRakhyani et al. [8] presented a WPT using MRC to optimize transfer efficiency and Q-factor for implanted medical devices. The system operated at a frequency of 700 KHz. The system included transmitter and receiver coils, and a 100 Ω load. Experimental results revealed that Q-factor optimization increased transfer efficiency by 100%. Transfer efficiency was 82% at 0.20 cm, whereas it was 72% at 0.32 cm. Advantages were safety for tissue. The disadvantage was the short distance. Dang et al. [9] designed and implemented a WPT system using MRC that optimized the efficiency of the source side to perform over a range of distances and to tolerate geometrical misalignment between the source and receiver. The operating frequency of the system was 2.75 MHz. The system consisted of a power source, receiver coil, and transmitting coil. To investigate the performance of the system, three configurations of transmitter source (i.e., series and shunt capacitor) were selected based on a control switch. Experimental results showed that a circuit containing a series coupling capacitor achieved better performance than one with a parallel coupling capacitor at short distances. The shunt coupling capacitor circuit achieved a longer transfer distance with lower transfer efficiency than with a series coupling capacitor. The system was tested over different transfer distances and several misalignment configurations. It achieved transfer efficiency of 93.1% at 3 cm and 19% at 14 cm. Advantages include tolerance of different distances, load characteristics, and alignment issues. The disadvantages was the large size.

Kyungmin et al. [10] designed and implemented a WPT using MRC for biomedical capsule endoscopy operating at 16.47 MHz oscillator frequency. The system consists of vision, control, data transmission components, and the MRC circuit. The vision system included a light emitting diode (LED). The system was intended for fast and accurate diagnosis. Data transmission included transmitter and receiver modules. MRC consisted of four coils: a power coil, transmitter coil, receiver coil, and load coil. Experimental results showed that transfer efficiency improved by 0.71% inside the patients compared to air at a distance between coils of 7 cm. The transfer efficiency was investigated relative to three parameters; distance, angle, and axial misalignment between the asymmetric transmitter and receiver coils. The authors concluded that the coupling coefficient was correlated with these parameters. Advantages included small size and high efficiency allowing use in small intestine endoscopy, gastroscopy, and colonoscopy. The disadvantage was a short coupling distance. Mohamadi et al. [11] implemented a WPT using zero voltage switching (ZVS) to operate in an MRC to use for implanted medical devices such as drug infusion, left ventricular assist devices and artificial hearts. The frequency control method was adapted to adjust the working frequency of the WPT system. The system consisted of the Proteus program, source coil, and load coil, control switch, resonator capacitor and advance virtual risc (AVR) microcontroller to control the voltage of the load circuit. Experimental results revealed that the voltage in the secondary coil decreased the coupling factor between the source and receiver coils. The output power of the secondary circuit was 1.5 W. Advantages were a long battery life and optimization efficiency. The disadvantage was size.

Zhang et al. [12] implemented a WPT system using MRC for medical sensors and implantable devices. The system consisted of an RF power source and an inductively coupled load. The operating frequency of the system was 7 MHz. The system was implanted within the abdomen about 3.5 cm under the skin of a Yorkshire pig. The maximum transfer distance between transmitter and receiver coils was about 10 cm with transfer efficiency of less than 22.3%. Advantages of the system were that it could be implanted and had a low cost. Disadvantages were its large in size, shape with an unsteady structure, and leakage of electric fields. Monti et al. [13] presented a WPT using MRC to supply implantable medical devices with power. The system operated at a frequency of 403 MHz. The system consisted of transmitter coil, receiver coil, rectifier circuit, and PM device. Maximum transfer efficiency of 5.24% was achieved between transmitter and receiver coils at a distance of 10 cm. The advantage was low cost. The disadvantage was that it was somewhat large. Campi et al. [14] implemented a WPT using an MRC for implantable medical devices. The operating frequency of the system was 300 KHz. The system consisted of a transmitting coil, a receiving coil, and a load. Experimental results revealed that transferred power in the receiving coil was 1 W. The transfer efficiency was almost 47% at 3 cm, whereas transfer efficiency was about 7% at 6 cm. The advantage was that it was a wearable device. The disadvantage was the short distance.

Gore et al. [15] implemented a WPT using MRC for implantable medical devices such as pacemakers to recharge the battery under load. The system operated at two different frequencies, 300 kHz and 13.56 MHz. The system consisted of a transmit coil, receive coil, rectifier, microcontroller (16F877A), ZigBee, relay, liquid-crystal display, and battery. The proposed system fully charged the battery then used ZigBee to send the battery status to the transmitter side and the relay was used to turn off wireless power transfer. The system displayed the received voltage and the battery voltage on an LCD. The maximum voltage transfer by the device was 11.05 V and was obtained at 15 cm. However, the system could be used over a maximum transfer distance of 30 cm. The output power of the secondary circuit was 400 μW. Advantages included safety for tissue. Disadvantages included high loss between coils. Campi et al. [16] carried out WPT using an MRC for recharging the battery of an implantable medical device such as pacemakers. The system operated at two different frequencies, 300 kHz and 13.56 MHz. The system consisted of a source coil, load coil, battery, and load. Two configurations were tested in their experiments. Experimental results revealed that transfer efficiencies for series-primary coil; series-secondary coil (SS) and a series-primary coil; parallel-secondary coil (SP) at 13.56 MHz were greater than at 300 KHz. The transfer efficiency of the SP configuration resulted in better performance at 300 KHz of about 31% at 3 cm compared with 13.56 MHz which showed 27% transfer at the same distance. The transfer efficiency of the SS configuration was better at 13.56 MHz at about 10% at 5 cm compared with 300 kHz which was 1% at the same distance. The output power of the secondary circuit was 1 W. The advantage was safety for tissue. Disadvantages were short distance and the environmental impact of batteries and their limited number of cycles.

Campi et al. [17] presented a WPT using an MRC for recharging the battery of an implanted pacemaker. The system operated at a frequency of 20 kHz. The system included a hardware DC power supply, DC to AC inverter, transmitting coil, receiving coil, and pacemaker device and used the software Multiphysics analysis program COMSOL. Two type of wire and three configurations of receiving coil were tested in their experiments. They used wires of 18 and 24 American Wire Gauge (AWG) in the receiving coil placed inside the pacemaker, placed outside the pacemaker, and integrated into the pacemaker for different numbers of turns. That result showed that transfer efficiencies and temperature increases for the receiving coil placed inside the pacemaker, placed outside the pacemaker, and integrated into the pacemaker were 41%, 77%, and 69%. Whereas temperature rises were 0.57, 0.16, and 0.14 °C. Advantages included safety for tissue. Disadvantages were not highlighted for the final configuration. Yeon et al. [18] designed and implemented a WPT to optimize the efficiency of MRC for brain regions. The operating frequency of the system was 137 MHz. The system consists of the transmitter coil, receiver coil, and load. Copper and gold coils were tested in their experiments. Experimental results showed that transfer efficiency and power at 2.8 cm in air was 0.76% and 240 μW, whereas 0.6% and 191 μW was obtained inside a lamb’s head, respectively. Advantages included not using a battery. Disadvantages were the short effective distance. Jawad et al. [19] designed and implemented a WPT using an MRC to use for home appliance operation such as mobile devices and charging batteries. The system consisted of an oscillator, transmitter coil, receiver coil, rectifier, DC-DC converter, and load. The system operated at a frequency of 1 MHz. Two types of coils were tested in their experiments: a single-tube loop coil (STLC) and a multi-turn copper wire coil (MTCWC). Experimental results revealed the transfer efficiency and power for STLC were better than MTCWC for a 100 Ω resistive load. Transfer efficiencies for using STLC were 80.66% and 66.66%. The transfer power using STLC was 4.84 and 4 W at 2 and 6 cm, respectively. Advantages included no battery requirement. Disadvantages included the large system and modest transfer efficiency.

Here, we obtained a high power based on an MRC system to operate a measurement unit for adequate power at different distances. The transfer power and efficiency are improved compared with that of previous work. The system used an Arduino Nano microcontroller, heart rate sensor, and nRF24L01 wireless protocol with adequate power at different distances. Data was exchanged between the measurement and monitoring units using the wireless protocol.

## 3. System Model

Our proposed MRC system consists of three units: power, measurement, and monitoring. The power unit included two subsystems: (i) a transmitter comprising an oscillator, a power supply, and a transmitter coil (Tx_C_) and (ii) a receiver comprised of a receiver coil (Rx_C_), a capacitor resonator, a bridge rectifier, and a regulator. The measurement unit consisted of an Arduino Nano microcontroller, a heart rate sensor, and an nRF24L01 module. The monitoring unit included an Arduino Mega 2560 microcontroller, an nRF24L01 module, and a laptop supported by serial monitoring to display the heart rate data. MRC operation was based on the distance between the transmitter and receiver coils illustrated in Figure 1. The power unit comprised of a transmitter coil (Tx_C_), receiver coil (Rx_C_), capacitor resonator, bridge rectifier, and regulator. The Tx_C_ converts an electric field to a magnetic field. There were two categories of coils used in this study, spiral and spider coils.

The bridge rectifier converts AC to DC using Schottky diodes (part SR260) [20] selected because of the low voltage drop, rated at 0.18 V, to ensure the maximum voltage was delivered. A 7805 voltage regulator maintained a voltage output from the rectifier of +5 V, the output voltage from the regulator supplied power to the Arduino Nano microcontroller programmed in C++ [21,22,23,24,25]. A voltage regulator was required because the output from the Rx_C_ depends on distance, so at short distances high voltages would damage the measurement unit.

## 4. Coils Design

The coils were wrapped manually using some available raw materials. Two types of coils spiral and spider coils were designed and constructed. The coils are described below: fundamentals of winding the spiral coil: the coil was wound around a screw mounted on a wooden plate. The coil was then bonded using silicon glue. The COMSOL 5.3 program was used for drawing a spiral coil. Figure 2a, illustrates the geometry of the coil.

Fundamentals of winding the spider coil: nine isosceles triangles with a 40° angle at the apex were machined using a computer numerically controlled (CNC) router. These triangles were arranged in a polygon and clamped in place with a screw and nut as shown in Figure 2b. The head of wire is wound from gap A1 to gap B1 from the top layer. Next, from gap B1 to gap C1 from the bottom layer. Then, from gap C1 to gap D1 from the top layer and then continues in this pattern. Figure 2b illustrates the method of wrapping spider coils the black line is on the top layer, while the dashed black is on the bottom layer. The plate has 1, 3, 5, 7, and 9 in the top layer, whereas 2, 4, 6, and 8 are on the bottom layer. When the coil was finished, the screw and nut were removed. Then, all triangles were pushed out of the coils. For the received coils, these coils are a prototype for which the number and the distance between the turns can be regulated by using a specific machine or using printed coils in future applications to reduce the size and weight for the purpose of implanting inside the body.

The properties of each of the coils is described in Table 1 and Table 2 and illustrated in Figure 3. The two categories of Tx_C_ and Rx_C_ are used in this paper are spiral and spider coils. The copper wire in the Tx_C_ used 21 AWG. A 21 AWG was used in the Rx_C_ spiral type and 23 AWG was used in the Rx_C_ spider type. WPT is based on coupling between the transmitting and receiving side and is obtained as a result of the magnetic flux caused by Tx_C_ current which would normally be weak. Tuning the coupling is required to achieve magnetic resonance which is done using capacitors on both sides. Magnetic resonance achieves the greatest magnetic flux and the greatest energy transfer on the receiving side. In this process, coils are used with different shapes and sizes depending on each application. Typically, the Tx_C_ is larger than the Rx_C_ in size and weight [26]. The value of the capacitors added to both sides depend on coil properties according to the Equation (1) [27].
(1)$$F_{r}=\frac{1}{2\pi \sqrt{L_{\mathrm{Tx}}C_{\mathrm{Tx}}}}=\frac{1}{2\pi \sqrt{L_{\mathrm{Rx}}C_{\mathrm{Rx}}}}$$

Electric current generates a magnetic field that can be envisaged as circular around the axis of the wire. The magnetic field at a radial distance can be calculated when passing current through a wire based on Equation (2) for a long straight wire according to the right-hand rule. This rule explains the direction of the current based on the right-hand rule, where the direction of the field is that of the fingers furl [28,29,30]. The total magnetic field can be calculated for a wire based on current from the Biot–Savart law related to Equation (3) [31]. This section will explain two important processes: (i) self-induction and (ii) mutual induction.
(2)$$B=\frac{\mu_{O\;}I\;}{2\pi r}$$
(3)$$B=\frac{\mu_{O\;}I\;}{4\pi }\int^{\text{​}}\frac{{dL}_{e}\;\text{×}\overset{\text{ˇ}}{r}}{r^{2}}$$
where *B* is magnetic field, *I* is current through the wire, *μo* is permeability of free space 4π10^−7^ T⋅m/A, *Le* is a length of wire, and *r* is radius or shortest distance to the wire.

Self-Induction is a phenomenon that occurs when the electric current changes in a coil to produce an induced electromotive force across the coil. In other words, it is a ratio between the induced electromotive force against rate of change of current. Self-Induction is measured in Henry (H) based on Equation (4) [32].
(4)$${e=l}_{S}\;\frac{d_{i}}{d_{t}}$$
where *e* is electromotive force across the coil, *l_S_* is self-induction, and *(di/dt)* is rate of change of current.

Two coils located near each other experience mutual induction, which is the effect exploited in motors, generators, and transformers. Current in the first coil with a number of windings (*N*_1_), generates a magnetic field (*B*_1_). Since the second coil is close to the first coil, some of the magnetic field lines will move to the second coil, with number of windings (*N*_2_), this is mutual inductance (*M*). *ø_21_* refers to the magnetic flux through the second coil as a result of current flowing through the first coil, this will generate an electromotive force (*e*_21_) due to the magnetic flow in the second coil according to Equation (5) [33]. Two coils show mutual induction between coils based on length of coil (*Le*), number of turns (*N*) in coils, and a cross-sectional area (*A*) of the coils according to Equation (6) [33].
(5)$$e_{21}{=-\;N}_{2}\;\frac{{d\text{ø}}_{\mathrm{21}}}{d_{t}}$$
(6)$$M=\frac{\mu_{0}N_{1}N_{2}A}{L_{e}}$$

## 5. Experiment Configuration of MRC

The proposed prototype system had three subsystems: power, measurement, and monitoring. The power unit involves two sections: the transmitter and receiver. The transmitter section includes the Tx_C_ and oscillator. Three types of coil topologies were used: spiral, spider, and a hybrid coil containing both spiral and spider, to explore transfer efficiency and transfer power between the transmitter and receiver side. The receiver section includes Rx_C_, capacitor resonator, bridge rectifier, voltage regulator, and load. The Rx_C_ experiences a changing magnetic field and converts it to an electric field. The voltage signal at the Rx_C_ was sinusoidal with a single pure tone, but its amplitude was lower than the amplitude of the transmitted voltage. The next stage used a capacitor resonator to achieve maximum power transfer between the two sides. The next stage convert the AC to DC with a bridge rectifier circuit then a 7805 voltage regulator maintains output voltage at +5 V. The output voltage from the regulator supplied power to the measurement unit. This measurement unit consists of an Arduino Nano microcontroller, a heart rate sensor and nRF24L01. The input voltage (VIN) pin on the Arduino Nano and ground pins were used to inject power. Three types of coils were used to transfer power between the transmitter and receiver. Compared to a direct connection (wire charging), the MRC, based on a transfer distance or airgap, has several advantages: (i) it does not require batteries used for power devices, (ii) the patient moves freely without being restricted by wires, (iii) it does not need a main source of electricity, therefore, it can be used anywhere, and (iv) the size of the coils receiver can be minimized to implant inside the human body to supply power to implantable devices.

Spiral–spiral topology: the output voltage signal of the TxC was sinusoidal shown in Figure 4. The detected signal on the RxC had the same frequency as the transmitted signal (i.e., 13.6 kHz) over a 7 cm air gap between transmitter and receiver coils. At the receiver circuit, the signal was converted from AC to DC by the bridge rectifier. The DC output voltage was 3.5 V over a 50 Ω resistive load. In addition, the maximum distance possible for correct function of the measurement unit was 9 cm with the voltage regulator removed from the circuit (to eliminate its intrinsic voltage drop), a sensor and the radio communication part (i.e., nRF24L01) were provided with 3.3 V voltage from Arduino Nano microcontroller shown in Figure 5. The voltage regulator was required across shorter distances of 1–5 cm to prevent potentially damaging high voltages. Figure 5, illustrates the topology. This system is a prototype and it experimented using air as transport medium. Where, the transfer power depends on magnetic field, self-induction and mutual induction between coils. Magnetic fields pass easily through most things for example tissues, bones, etc.

For the spider–spider topology, the output voltage signal of Tx_C_ was the sinusoid shown in Figure 6. The detected signal at the Rx_C_ had the same frequency as the transmitted signal (i.e., 13.6 kHz) over a 7 cm air gap between the Tx_C_ and Rx_C_. At the receiver circuit the signal was converted from AC to DC by the bridge rectifier. The DC output voltage of 18.5 V was observed at 7 cm over a 50 Ω resistive load. The maximum distance over which the spider–spider topology was able to power the measurement unit (Arduino Nano microcontroller, a heart rate sensor, and used the nRF24L01 wireless protocol) without the voltage regulator was 20 cm while using a regulator the circuit functioned between 1 and 18 cm. Figure 7 illustrates the coil topology used.

Spiral–spider topology: the output voltage signal over the Tx_C_ was sinusoidal. At the receiver circuit, the signal was converted from AC to DC voltage by the bridge rectifier. A DC output voltage of 20 V was measured at 7 cm over a 50 Ω resistive load. The spiral–spider topology operated the sensor system without failure at 20 cm without using the voltage regulator. Using the regulator the circuit would function between 1 to 18 cm. Figure 8 illustrates the third topology for supplying a measurement unit with power.

## 6. Result and Discussions

This subsection presents the voltage signal from the various coil topologies exploring transfer efficiency and output power at different distances, compared among topologies, and compared the present work with related previous studies by transfer power as well as efficiency at distance. Voltage and spectrum signals were captured using a storage oscilloscope model UTD2025CL over the Tx_C_ and Rx_C_. Measurement and correlation of transfer efficiency and output power delivered to resistive loads at different distances for each topology are presented. Finally, a comparison is made between the current study and previous work.

### 6.1. Spiral–Spiral Topology

#### 6.1.1. Analysis of the Voltage Signal under MRC

The output voltage signal of the transmitter, the receiver and the frequency spectrum were measured using a storage oscilloscope (UTD2025CL). The *y-axis* indicates the amplitude of the signal and the *x-axis* is time. The output voltage signal of the Txc was a sinusoidal signal and the frequency of the oscillator circuit was 13.6 kHz, shown in Figure 9a. At the receiver side, Rx_C_ indicated a signal from Tx_C_ at 7 cm with the same frequency as Tx_C_, illustrated in Figure 9b. Figure 9a,b shows the frequency spectrum (red curve) of the transmitter and receiver circuit as a single tone without noise or mixed higher frequency signals. At the receiver circuit, the signal was converted from AC to DC by the bridge rectifier. Consequently, the DC output voltage of 3.5 V observed at 7 cm with 50 Ω RL, systems shown in Figure 9c.

At 9 cm, the output voltage signal of the Tx_C_ was sinusoidal and the frequency of the oscillator circuit was 13.8 kHz, illustrated in Figure 10a. The Rx_C_ detected a signal at 13.6 kHz, shown in Figure 10b. Figure 10a,b shows the frequency spectrum (red line) of the transmitter and receiver circuit as a single tone without noise or interference signals. At the receiver circuit, the signal was adapted from AC to DC voltage by the bridge rectifier. Consequently, the DC output voltage of 4.5 V recorded at 9 cm, shown in Figure 10c.

#### 6.1.2. Performance Metrics Evaluation

Figure 11a–c shows the main results from tests of the proposed MRC technique. To establish the relationship, the transfer distances in centimeters is plotted on the *x-axis* and the transfer efficiencies for DC and AC sources as a percentage are plotted on the *y-axis*. The transfer efficiency is determined according to Equation (7) [19].
(7)$$Transfer\;efficiency\;\%\;(ƞ)=\frac{output\;power}{input\;power}\times 100\%\;$$

Figure 11a shows the measured transfer efficiency at different loads (RL). The transfer distances in centimeters are plotted on the *x-axis*, and the transfer efficiencies at RL as a percentage are plotted on the *y-axis*. The transfer efficiency was 27.1%, 26.26%, 28.01%, and 13.54% at 5 cm. However, it gradually decreased at distances over 5 cm. In addition, the transfer efficiency was 2.32%, 3.09%, 2.36%, and 0.85% at 12 cm. whereas, the transfer efficiency at 20 cm was 0.14%, 0.16%, 0.14%, and 0.06% at 40, 50, 70, and 200 Ω, respectively. Figure 11b shows the measured transfer power at different resistive loads. The transfer distances in centimeters are plotted on the *x-axis* and the transfer power in watts are plotted on the *y-axis*. The transfer power was 3.79, 4.173, 5.136, and 5.27 W at 5 cm. However, it gradually decreased at distances over 5 cm. In addition, the transfer efficiency was 0.195, 0.2079, 0.22794, and 0.231 W at 12 cm, whereas the transfer efficiency at 20 cm was 0.00945, 0.0144, 0.0176, and 0.0154 W at 40, 50, 70, and 200 Ω, respectively. Figure 11c, indicates the measured transfer power on the Rxc through different resistive loads. The transfer distances in centimeters are plotted on the *x-axis*, and the transfer power on the Rxc at RL in percentage are plotted on the *y-axis*. The transfer power was 9.1, 13.3, 12.3, and 13 W at 5 cm. However, it gradually decreased at distances over 5 cm. In addition, the transfer efficiency was 0.56, 0.76, 0.84, and 1 W at 12 cm. at 40, 50, 70, and 200 Ω, respectively.

#### 6.1.3. Correlation between Performance Metrics

The Table 3 illustrates parameters indicate low transfer distances that achieve high transfer power and efficiency. High indicates high transfer distance with low transfer power and efficiency. The maximum transfer distance between the transmitter and receiver side was 25 cm. Resistive loads of 40, 50, and 70 Ω were used to measure transfer power and efficiency. Therefore, the amount of input/output power measured to determine transfer efficiency for the system at RL. In Table 3, the transfer power and efficiency were 1.386, 1.71, and 2.08 W as well as 19.8%, 20.02%, and 21.14% at 7 cm, respectively. These values gradually decreased at a distances of over 7 cm. In addition, the transfer power and efficiency were 0.009, 0.014, and 0.017 W as well as 0.14%, 0.16%, and 0.14% at 20 cm, respectively.

### 6.2. Spider–Spider Topology

#### 6.2.1. Voltage Signal Based on MRC

The output voltage signal of the Tx_C_ was a sinusoid at the frequency of the oscillator circuit of 13.7 kHz, illustrated in Figure 12a. At the receiver side, Rx_C_ detected a signal from Tx_C_ over an air gap of 7 cm with the same frequency as in Tx_C_, shown in Figure 12b. Figure 12a,b show the frequency spectrum (red line) of the transmitter and receiver circuit as a single tone without noise or interference signals. At the receiver circuit, the bridge rectifier converted the signal from AC to DC. Consequently, the DC output voltage of 18.5 V recorded at 7 cm over a 50 Ω resistive load.

For the second topology to supply the measurement unit at 20 cm, the output voltage signal of the Tx_C_ was a sinusoid and the frequency of the oscillator circuit was 13.6 kHz, illustrated in Figure 13a. At the receiver circuit, Rxc detected a signal from the Tx_C_ over an air gap of 20 cm at the same frequency, illustrated in Figure 13b. Figure 13a,b shows the frequency spectrum (red line) of the transmitter and receiver circuit as a single tone without noise or interference signals. At the receiver circuit, the bridge rectifier converted the signal from AC to DC. Consequently, the DC output voltage of 4.6 V recorded at 20 cm, shown in Figure 13c.

#### 6.2.2. Performance Metrics Evaluation

Second topology, Figure 14a, indicates the measured transfer efficiency at different loads (RL). The transfer distances in centimeters are plotted on the *x-axis*, and the transfer efficiencies in percentage are plotted on the *y-axis*. The transfer efficiency in a 300 Ω load was large compared with other resistive loads at distances from 1–9 cm, ranging from 96.8% to 86%. Whereas the transfer efficiency for 50 Ω was low compared to other loads at distances 1–5 cm, ranging from 61.12%, 54.84%, 40.33%, 38.18% and 33.11%. Transfer efficiency into 400 Ω was high compared with other loads at distances 10–25 cm, ranging from 75.7%, 60.9%, 43.9%, 22.8%, 14.7%, 8.24%, 5.39%, 3.41%,2.13%, 1.39%, 0.82%, 0.72%, 0.49%, 0.29%, 0.26% and 0.23%. Transfer efficiency in 25 Ω was low compared with other loads at distances 6–25 cm, ranging from 17.95%, 8.33%, 5.58%, 2.23%, 0.7%, 0.37%, 0.18%, 0.11%, 0.04%, 0.04%, 0.03%, 0.03%, 0.02%, 0.02%, 0.02%, 0.02%, 0.01%, 0.01%, 0.01% and 0.01%. The remaining values fell between these values. Figure 14b, indicates the measured transfer power at different resistive loads. The transfer distances in centimeters is plotted on the *x-axis* and the transfer power in watts plotted on the *y-axis*. The transfer power for 400 Ω was high compared with other resistive loads at distances from 1 to 25 cm, which ranged from 13, 12, 10, 8.7, 7.2, 5.5, 4, 3.6, 2.8, 2, 1.2, 0.9, 0.5, 0.4, 0.3, 0.2, 0.2, 0.1, 0.1, 0.1, 0.1, 0.1, 0.1, 0.1, 0.1 W. The transfer power into 25 Ω was low compared to other loads at distances from 1 to 25 cm, which ranged from 7.5, 3.653, 2.08, 1.68, 0.95, 0.616, 0.3565, 0.23, 0.15, 0.10168, 0.07038, 0.04704, 0.036, 0.0248, 0.0182, 0.0132, 0.00969, 0.00748, 0.006, 0.00442, 0.0033, 0.0026, 0.00207, 0.00168 and 0.00133 W. The remaining values fall between these values. 

Figure 14c, shows the measured transfer power on the RxC into different resistive loads. The transfer distance in centimeters is plotted on the x-axis, and the transfer power in watts is plotted on the y-axis. The transfer power into 100 Ω was high compared with other resistive loads at distances 1–3 cm, ranging from 126, 121.52 and 103.2 W. The transfer power into 50 Ω was low compared with other resistive loads at distances 1–5 cm, ranging from 58, 45.24, 29.04, 25.2 and 18.5 W. The transfer power into 200 Ω was high compared with other resistive loads at distances 4–8 cm ranging from 96.05, 90.2, 76.845, 69.3 and 59 W. Transfer power into 25 Ω was low compared with other resistive loads at distances 6–25 cm, ranging from 14, 6, 4, 1.6, 0.504, 0.265, 0.08, 0.03, 0.026, 0.022, 0.02, 0.016, 0.014, 0.012, 0.011, 0.09, 0.08, 0.07 and 0.06 W. The transfer power into 400 Ω was high compared with other resistive loads at distances 10–25 cm, ranging from 46.64, 37.5, 27, 14, 9, 5.06, 3.3, 2.09, 1.304, 0.852, 0.504, 0.444, 0.3, 0.18, 0.16, 0.14 W. Several resistive loads used to take into account the type of pregnancy or the device used in the future. The output power depends on the amount of voltage and current for each resistive load, the output voltage depends on the amount of RL and the transfer distance between coils. Which in turn affects the transfer efficiency for each distance.

#### 6.2.3. Correlation between Performance Metrics

The Table 4 illustrates parameters: The indicated low transfer distances are with a high transfer power and efficiency. High indicates high transfer distances with low transfer power and efficiency. The maximum transfer distance between the transmitter and receiver side was 25 cm. Various loads were used, 25, 33, 50, 100, 200, 300, and 400 Ω to measure transfer power and efficiency over distance. Therefore, the amount of input to output power was determined to measure transfer efficiency for the system at RL. Table 4, illustrates transfer power and efficiency at distance. The transfer power and efficiency were 0.95, 2.13, 3.04, 4.8, 6.75, 6.6, and 7.2 W as well as 31.79%, 36.51%, 33.11%, 76.45%, 93.37%, 94.5%, and 92.1% at 5 cm, respectively. Efficiency gradually decreased at distances over 5 cm.

### 6.3. Spiral–Spider Topology

#### 6.3.1. Voltage Signal Based on MRC

The output voltage signal of the Tx_C_ was sinusoidal and the frequency of the oscillator circuit was 13.6 kHz, illustrated in Figure 15a. At the receiver side, Rx_C_ detected a signal from Tx_C_ over an air gap of 7 cm with the same frequency for Tx_C_, illustrated Figure 15b. Figure 15a,b shows the frequency spectrum (red line) of the transmitter and receiver circuit as a single tone without noise or interference signals. At the receiver circuit, the signal was converted from AC to DC by the bridge rectifier. Consequently, the DC output voltage was 20 V recorded at 7 cm over a 50 Ω resistive load.

Spiral–spider topology for supplying a measurement unit at 20 cm, the output voltage signal of the Tx_C_ was a sinusoid and the frequency of the oscillator circuit was 13.7 kHz, illustrated in Figure 16a. At the receiver circuit, Rx_C_ detected a signal from the Txc over an air gap of 20 cm at 13.7 kHz frequency, illustrated in Figure 16b. Figure 16a,b shows the frequency spectrum (red line) of the transmitter and receiver circuit as a single tone without noise or interference signals. At the receiver circuit, the signal was converted from AC to DC by the bridge rectifier. Consequently, the DC output of 5 V recorded at 20 cm, shown in Figure 16c.

#### 6.3.2. Performance Metrics Evaluation

Figure 17a, indicates the measured transfer efficiency at RL. The transfer distance in centimeters is plotted on the *x-axis* and the transfer efficiency in percentage plotted on the *y-axis*. The transfer efficiency into 100 Ω was high compared with other resistive loads at distances 1–2 cm, ranging from 99.03% and 97.6%. The transfer efficiency into 300 Ω was high compared with other resistive loads at distances 3–7 cm, which ranged from 95.07%, 95%, 94.93%, 91.55% and 86.63%. The transfer efficiency into a 400 Ω load was high compared with other resistive loads at distances 8–25 cm, which ranged from 87.84%, 86.27%, 80%, 71.37%, 50.4%, 40%, 27.2%, 16.2%, 9.2%, 4%, 2.88%, 1.8%, 1.3%, 0.96%, 0.6%, 0.54%, 0.32% and 0.28%. Furthermore, the transfer efficiency into a 25 Ω load was low compared to other resistive loads at distances 1–25 cm, ranging from 92.44%, 85.03%, 82.55%, 53.26%, 40.33%, 29.41%, 17.54%, 9.07%, 3.73%, 2.13%, 1%, 0.53%, 0.28%, 0.27%, 0.22%, 0.18%, 0.18%, 0.13%, 0.06%, 0.05%, 0.05%, 0.04%, 0.04%, 0.04% and 0.03%. The remainder of the values fell between these values. Figure 17b, indicates the measured transfer power at different resistive loads. The transfer distance in centimeters is plotted on the *x-axis*, and the transfer power at RL in percent is plotted on the *y-axis*. The transfer power into a 50 Ω load was high compared with other resistive loads at distances 1–4 cm, ranging from 46.41, 38.07, 29.6 and 23.45 W. Whereas the transfer power into 400 Ω was low compared with other resistive loads at distances 1–5 cm, which ranged from 5.04, 4.18, 4.1, 4.73 and 4.84 W. The transfer power for a 100 Ω load was high compared with other resistive loads at distances 5–6 cm, ranging from 15.8 and 10.4 W. The transfer power into a 25 Ω load was low compared with other loads at distances 6–25 cm, which ranged from 4, 2.48, 1.5325, 1.05, 0.672, 0.455, 0.345, 0.198, 0.15, 0.0915, 0.068, 0.0495, 0.04, 0.025, 0.021, 0.0132, 0.009, 0.007 and 0.005 W. The rest of the values fell between these values.

Figure 17c, indicates the measured transfer power on the Rx_C_ at different resistive loads. The transfer distance in centimeters is plotted on the *x-axis* and the transfer power on the Rx_C_ at RL in percent is plotted on the *y-axis*. The transfer power into a 50 Ω load was high compared with other resistive loads at distances 1–3 cm ranging from 112, 93.3 and 81.6 W. The transfer power into 400 Ω is low compared with other resistive loads at distances 1–2 cm, ranging from 51.2 and 49.6 W. The transfer power for a 200 Ω load was high compared with other resistive loads at distances 4–8 cm, which ranged from 69.3, 72, 73.8, 69.3 and 59.4 W. The transfer power into a 400 Ω load was high compared with other resistive loads at distances 10–25 cm, ranging from 40.8, 36.4, 25.2, 20, 13.6, 8.1, 4.6, 2, 1.44, 0.9, 0.65, 0.48, 0.3, 0.27, 0.16 and 0.14 W. While the transfer power into a 25 Ω load was low compared with other resistive loads at distances 3–25 cm, ranging from 44, 24.5, 18.15, 13.5, 8.05, 4.08, 1.68, 0.96, 0.45, 0.24, 0.128, 0.12, 0.1, 0.08, 0.08, 0.06, 0.027, 0.024, 0.021, 0.02, 0.019, 0.016 and 0.014W. The rest of the values fell between these values.

#### 6.3.3. Correlation between Performance Metrics

The Table 5 illustrates parameters, indicate low transfer distances achieving a high value for transfer power and efficiency. High indicates high transfer distances achieving low transfer power and efficiency. The maximum transfer distance between the transmitter and receiver side was 25 cm. We used various loads 25, 33, 50, 100, 200, 300, and 400 Ω to measure transfer power and efficiency at distance. The amount of input vs output power was determined to measure transfer efficiency for the system at RL. The transfer power and efficiency were 5.4, 10.08, 12.5, 16, 10, 7, and 5 W as well as 40.33%, 65.45%, 83.04%, 76.05%, 87.27%, 94.93%, and 89.23% at 5 cm, respectively. Both measures gradually decreased at distances over 5 cm.

## 7. Comparison among Topologies

This subsection presents a comparison among topologies at RL of transfer power and efficiency. We compare topologies into loads of 40 and 200 Ω. The transfer distance in centimeters is plotted on the *x-axis* and the transfer power at RL in watts is plotted on the *y-axis*. Figure 18a,b, indicates the measured transfer power at RL. Figure 18a illustrates that the transfer power for the spiral–spider topology is the best other topology according to transfer power into a 40 Ω load. Transfer power at 5 cm was 10.08, 3.12, and 2.136 W for spiral–spider, spiral–spiral, and spider–spider, respectively. While, it was 1, 0.3, and 0.4 W at 10 cm for spiral–spider, spiral–spiral, and spider–spider, respectively. Figure 18b, indicates the measured transfer power at RL, the transfer power for the spiral–spider topology is the best other topology at 200 Ω. The spiral–spider is higher efficiency than other topologies at 3–25 cm. This Figure illustrates transfer of power into a 200 Ω load, it was 22, 17.05, and 12.6 W at 1 cm for spiral–spiral, spiral–spider, and spider–spider, respectively. In addition, it was 9.9, 6.75, and 5.27 W at 5 cm. Furthermore, it was 2, 1.2, and 0.5 W at 5 cm for spiral–spider, spider–spider, and spiral–spiral, respectively.

Figure 19a,b, indicates the measured transfer efficiency into 40 and 200 Ω. Transfer distance in centimeters is plotted on the *x-axis*, and the transfer efficiency at 40 Ω load in percent is plotted on the *y-axis*. Figure 19a, indicates the measured transfer efficiency into 40 Ω. The spiral–spider topology is the best compared with other topologies at 1–25 cm, it was 93%, 65.45%, and 2% at 3, 5, and 11 cm, respectively. Figure 19b, indicates the measured transfer efficiency at 200 Ω. The spiral–spider topology is the best compared with another topology at 1–25 cm, it was 93%, 65.45%, and 2% at 3, 5, and 11 cm, respectively. Figure 19b, indicates the measured transfer efficiency at 200 Ω. The spider–spider topology is best compared with other topologies at 1–8 cm, it was 96% and 94% at 3 and 5 cm, respectively. The spiral–spider topology is the best compared with other topologies at 9–25 cm, it was 55% and 4% at 10 and 15 cm, respectively. The spiral–spiral topology is the lower transfer efficiency at 40 and 200 Ω compared with other topologies at 1–25 cm. There are several reasons for this in terms of low transfer power and the amount of the coil value for both the transmitter and receiver.

The transfer distance in centimeters is plotted on the *x-axis*, and the transfer power on RxC at RL in watts plotted on the *y-axis*. Figure 20a,b, indicates the measured transfer power on RxC into 40 and 200 Ω. Figure 20a, indicates the measured transfer power on RxC into 40 Ω. The spider–spider topology is the best compared with other topologies at 1–4 cm, it was 68.2 W at 3 cm. Also, the spiral–spider topology is effective compared with the other topologies at 5–25 cm, it was 36 and 0.2 W at 5 and 15 cm, respectively. Figure 20b, indicates the measured transfer power on RxC into a 200 Ω load, the spider–spider topology performed the best compared with other topologies at 1–6 cm, it was 90.2 and 111 W at 2 and 5 cm, respectively. Transfer power for spider–spider and spiral–spider topologies were almost equal at 7–25 cm, it was 30.8 and 30 at 10 cm for these topologies, respectively. Spiral–spider behavior this shape for a specific reason is that the amount of transfer power between coils is low due to the extra load on the oscillator, which affects the low energy entering the oscillator and affects the value of self-induction and mutual induction.

## 8. Comparison with Previous Work

The table compares the performance metrics of the previous studies in terms of transfer power, efficiency and distance concerning the present study. Considering transfer power of previous work: we found transfer power of 11 W at 2 cm, whereas it was 4.48 W in [19], with a transfer power of 8.4 W at 3 cm, it was 0.00024 W in [18]. In addition, transfer power was 5.332 W at 6 cm, while it was 1 W in [14] and 4 W in [19]. Transfer power was 1.87 W at 10 cm and 0.001 W in [13] and 0.0004 W in [15]. We found transfer efficiency was 95.5% at 3 cm, whereas it was 93.1% in [9], 0.76% in [18], and 76% in [19]. We found transfer efficiency of 94.48% at 6 cm, while it was 7% in [14] and 66% in [19]. Furthermore, the transfer efficiency was 93.37% at 7 cm, while it was 0.71% in [10] and 60% in [19]. We found transfer efficiency of 86.01% at 9 cm, with 22.3% reported in [12] and 57% in [19]. The transfer efficiency was 60.55% at 10 cm, while it was 5.24% in [13] and 55% in [19]. Table 6 summarizes a previous studies and compare their performance metrics in terms of objective, operating frequencies, implementation environment, application, transfer distance, transfer efficiency, and transfer power.

## 9. Conclusions

This paper introduces the design and implementation of an MRC for efficiently supplying power to medical devices such as heart rate sensors over substantial distances. The MRC system was tested on transfer distances of 1–25 cm with different topologies. The WPT system contained three units: power unit, measurement unit, and monitoring unit. The measurement unit for this system included a heart rate sensor, Arduino Nano microcontroller, and an nRF24L01 wireless protocol for heart rate data transmission. The WPT system was experimentally tested for different coil topology: spiral–spiral, spider–spider, and spiral–spider. The experiments were conducted at different transfer distances between the transmitter and receiver coils (borne by the patient). The first topology at 7 cm had 21.14% and 2.079 W transfer efficiency and power, respectively, into a 50 Ω load. We observed that the measurement unit of the system could function properly without failure at a distance of >9 cm for the first topology. However, we recommend using the system at distances of <9 cm to avoid system failure or damage. The second topology at 7 cm had 87.05% efficiency and 4 W power transfer, respectively, into a 400 Ω load. We observed that the measurement unit of the system could function properly at 20 cm with the second topology if the voltage regulator was removed to recover the intrinsic voltage drop of the device. At 7 cm, the third topology had 86.15% and 5 W transfer efficiency and power, respectively, at 400 Ω. We observed that the measurement unit of the system could function properly without failure at >20 cm with the third coil topology if the voltage regulator was removed to recover the intrinsic voltage drop of the device. Transfer power, transfer efficiency, transfer distance, and voltage generated by the proposed system were superior to that of previous studies. Future work will focus on design and implement a system with an integrated antenna in a printed spiral and spider coils to reduce the size and weight of the implantable device. In addition, a standalone microcontroller can be used to reduce the power consumption of the measurement system. Moreover, use of a simulation program to study the parameters and coil values for both those received and transmitted.

## Figures and Tables

**Figure 1 sensors-20-02549-f001:**
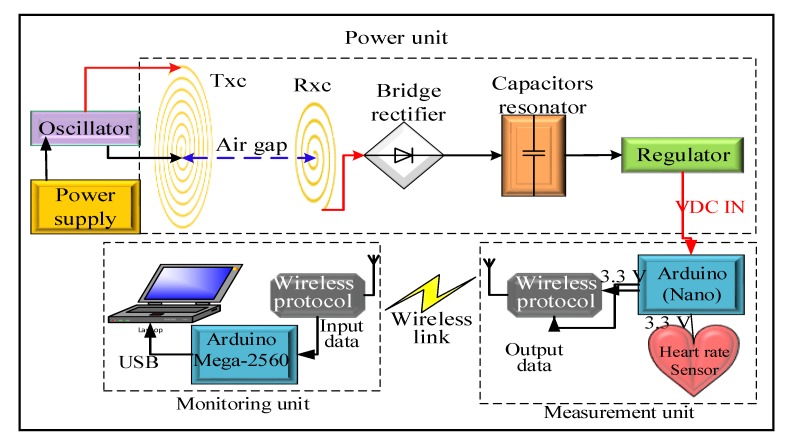
Block diagram of the use of a magnetic resonator coupling (MRC) for a heart rate sensor.

**Figure 2 sensors-20-02549-f002:**
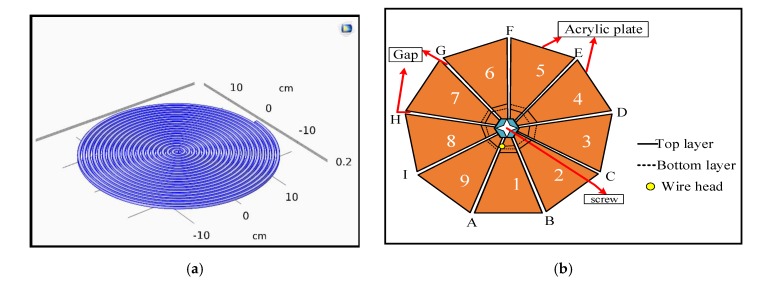
A method of wrapping (**a**) spiral coil and (**b**) spider coil.

**Figure 3 sensors-20-02549-f003:**
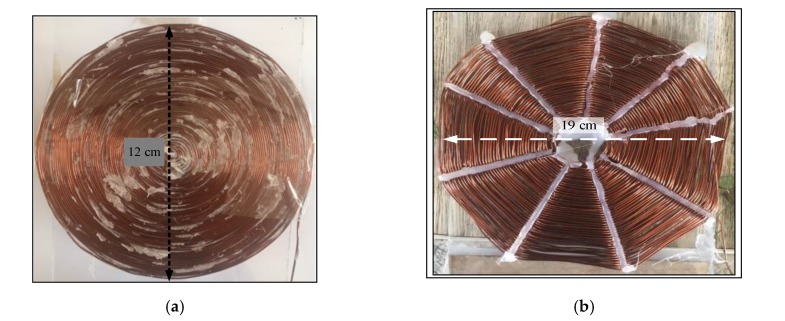
Types of coils (**a**) spiral coil and (**b**) spider coils.

**Figure 4 sensors-20-02549-f004:**
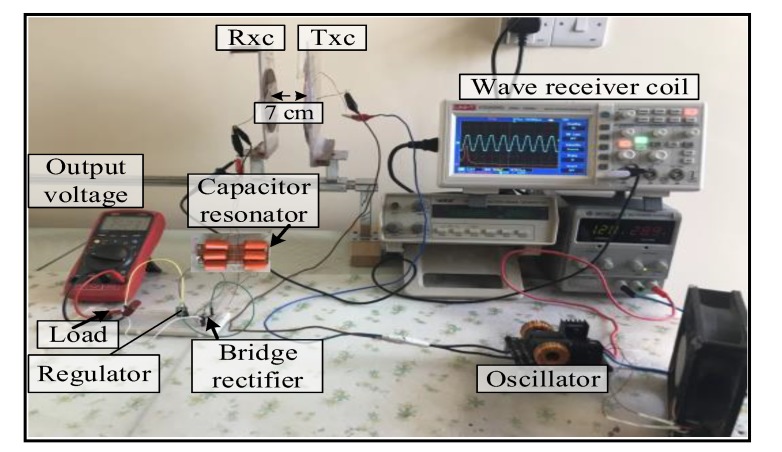
A snapshot of the MRC system at 7 cm for first topology at load.

**Figure 5 sensors-20-02549-f005:**
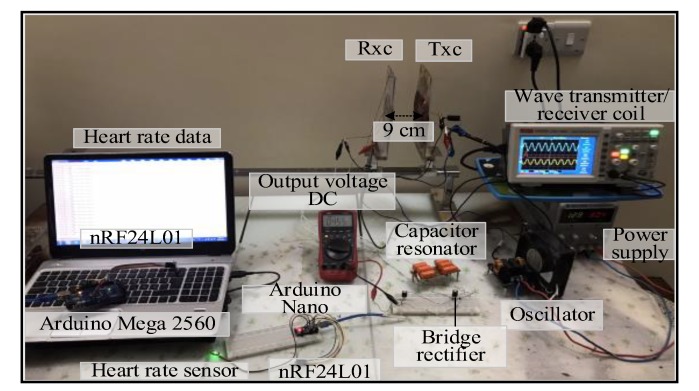
A snapshot of the MRC system at 9 cm for first topology for real time.

**Figure 6 sensors-20-02549-f006:**
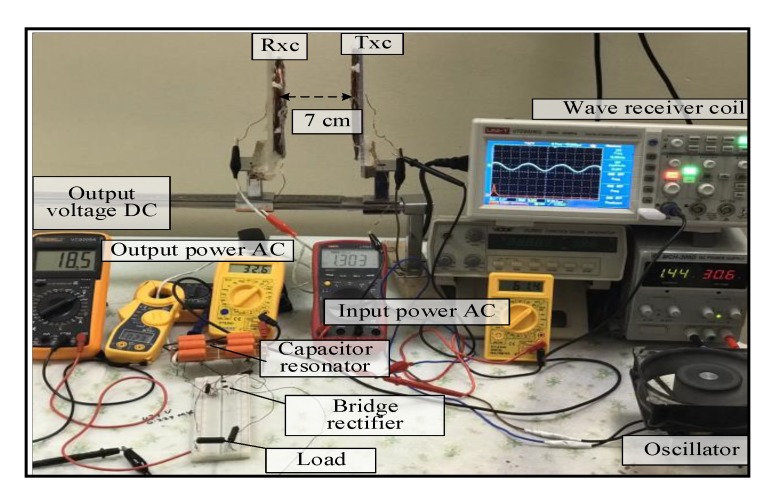
A snapshot of the MRC system at 7 cm for the second topology under load.

**Figure 7 sensors-20-02549-f007:**
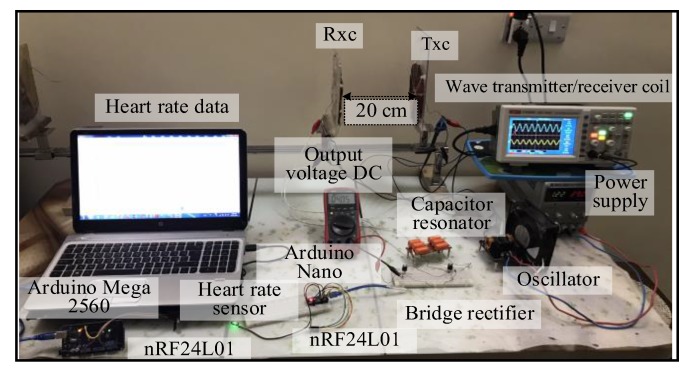
A snapshot of the MRC system at 20 cm for the second topology for real time.

**Figure 8 sensors-20-02549-f008:**
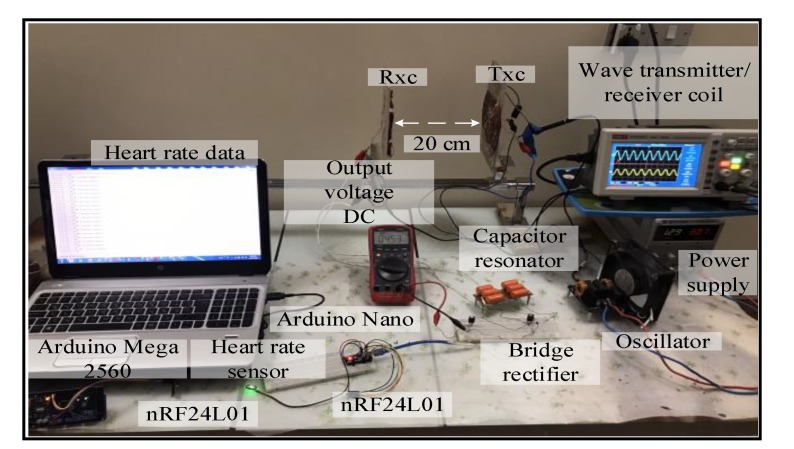
A snapshot of the MRC system at 20 cm for the third topology.

**Figure 9 sensors-20-02549-f009:**
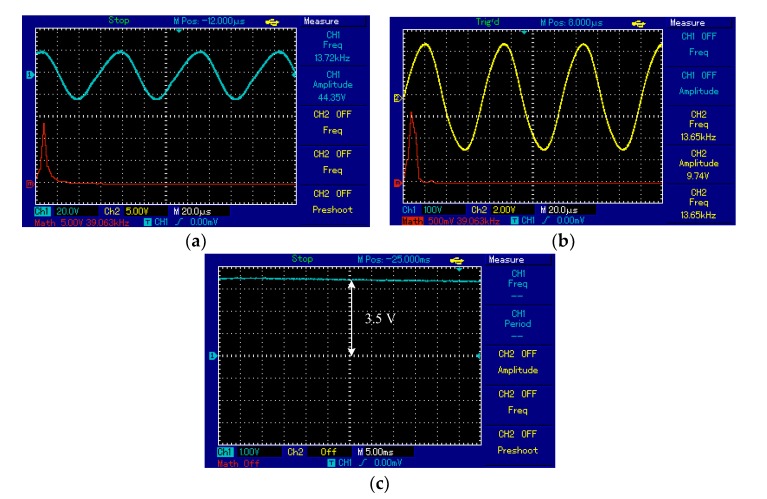
The voltage waveforms using the spiral–spiral topology at 7 cm for 50 Ω load: (**a**) transmitter coil, (**b**) receiver coil, and (**c**) voltage load.

**Figure 10 sensors-20-02549-f010:**
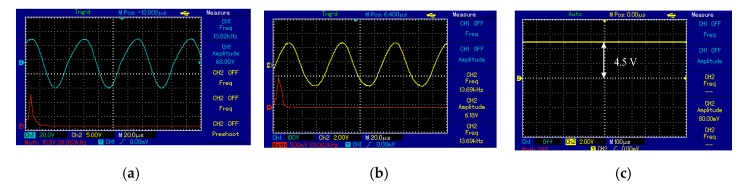
The voltage waveforms using the first topology at 9 cm for the measurement unit: (**a**) transmitter coil, (**b**) receiver coil, and (**c**) voltage load.

**Figure 11 sensors-20-02549-f011:**
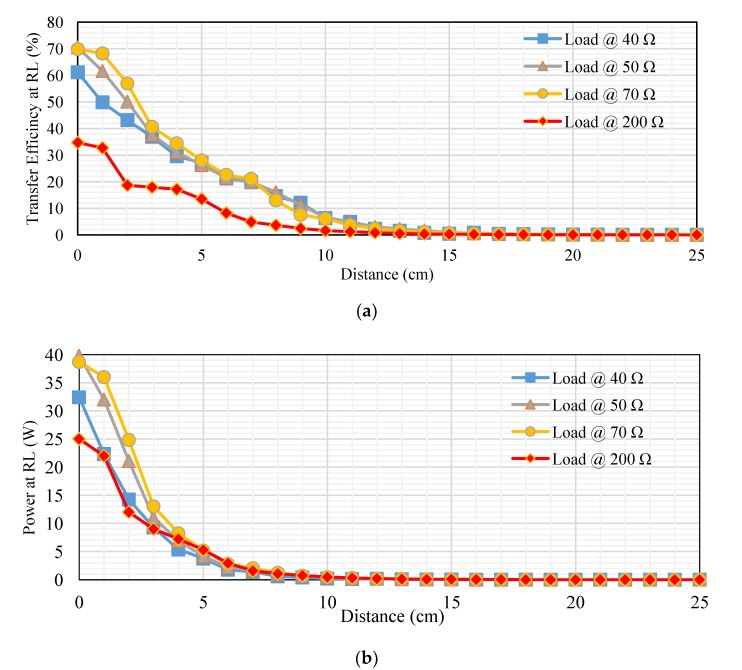
Performance at different loads for first topology: (**a**) transfer efficiency at different loads (RL), (**b**) transfer power at RL, and (**c**) transfer power on Rx_C_ at RL.

**Figure 12 sensors-20-02549-f012:**
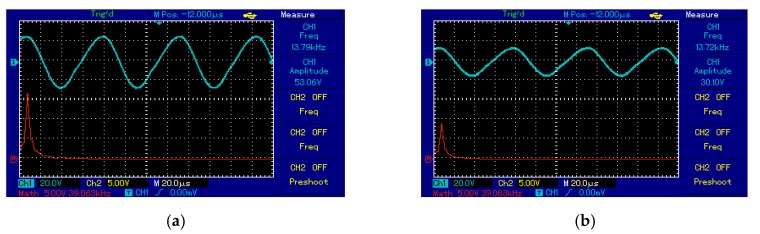
The voltage waveforms using second topology at 7 cm for: (**a**) transmitter coil and (**b**) receiver coil.

**Figure 13 sensors-20-02549-f013:**
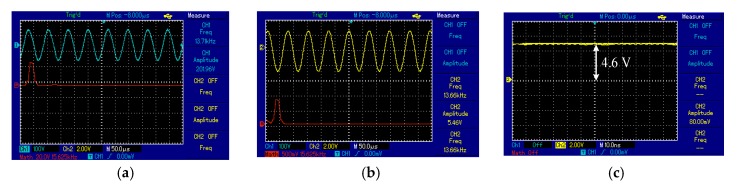
The voltage waveforms using second topology at 20 cm for measurement unit: (**a**) transmitter coil, (**b**) receiver coil, and (**c**) voltage load.

**Figure 14 sensors-20-02549-f014:**
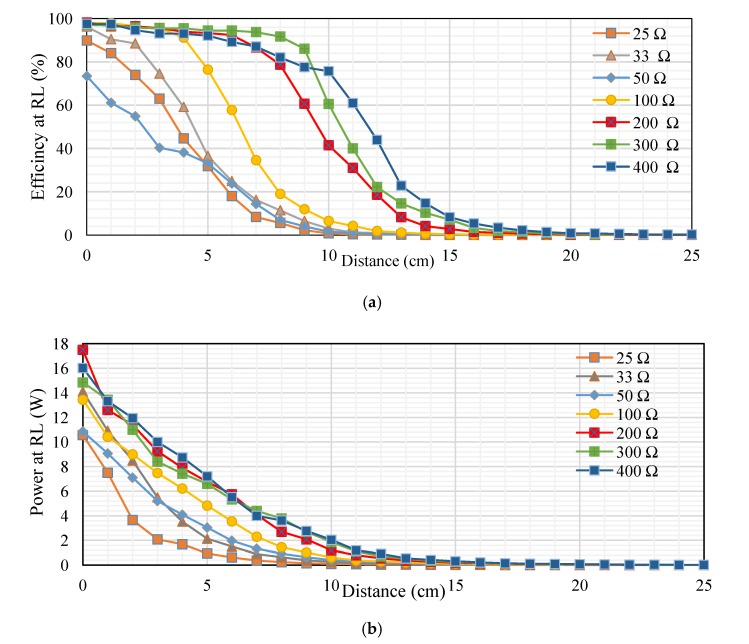
Performance at different loads for second topology: (**a**) transfer efficiency at RL, (**b**) transfer power at RL, and (**c**) transfer power on Rx_C_ at RL.

**Figure 15 sensors-20-02549-f015:**
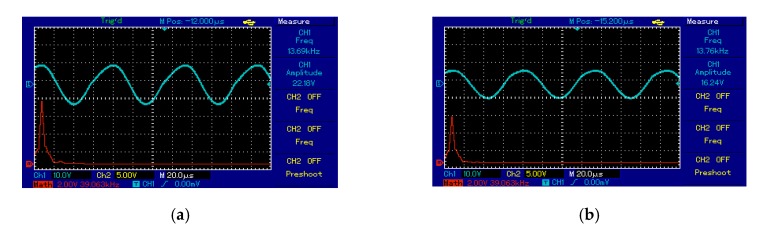
The voltage waveforms using third topology at 7 cm for: (**a**) transmitter coil and (**b**) receiver coil.

**Figure 16 sensors-20-02549-f016:**
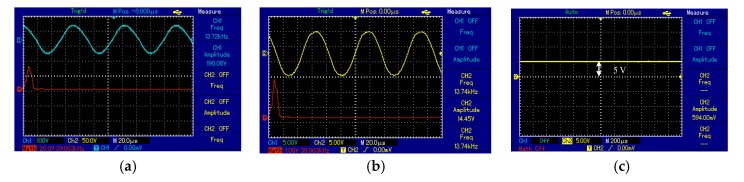
The voltage waveforms using the third topology at 20 cm for measurement unit: (**a**) transmitter coil, (**b**) receiver coil, and (**c**) voltage load.

**Figure 17 sensors-20-02549-f017:**
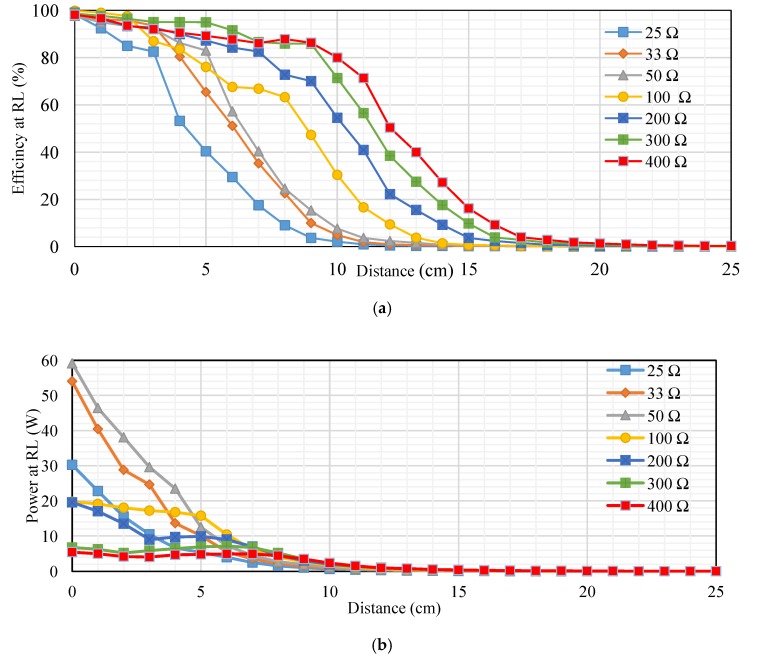
Performance at different loads for third topology: (**a**) transfer efficiency at RL, (**b**) transfer power at RL, and (**c**) transfer efficiency on Rx_C_ at RL.

**Figure 18 sensors-20-02549-f018:**
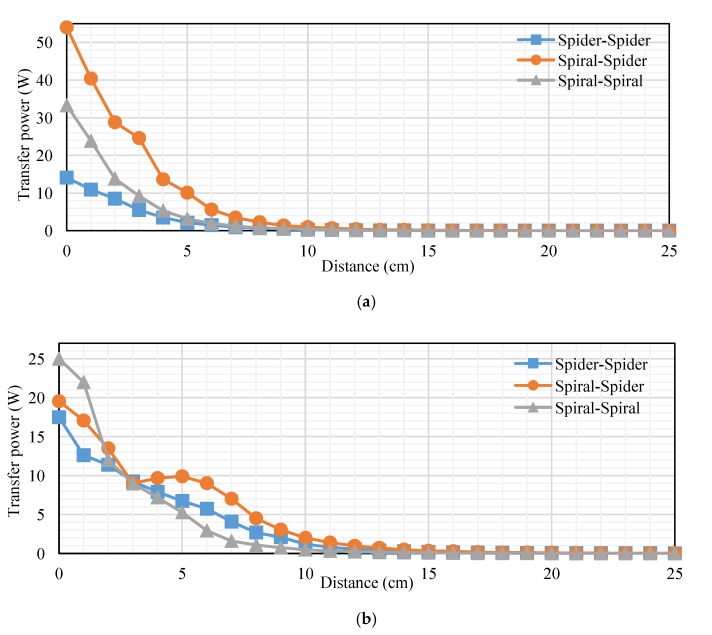
Transfer power at RL based-topologies at: (**a**) 40 Ω and (**b**) 200 Ω.

**Figure 19 sensors-20-02549-f019:**
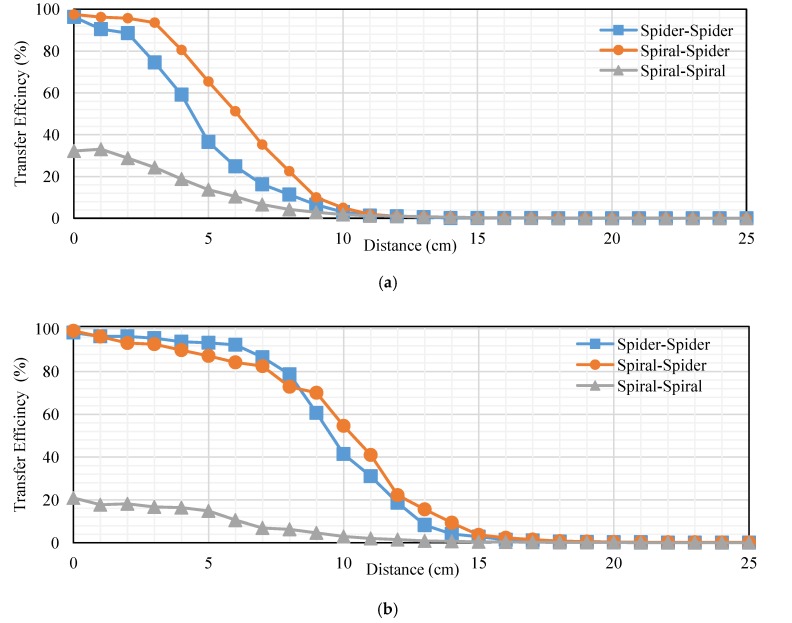
Transfer efficiency of RL based topologies into: (**a**) 40 Ω and (**b**) 200 Ω.

**Figure 20 sensors-20-02549-f020:**
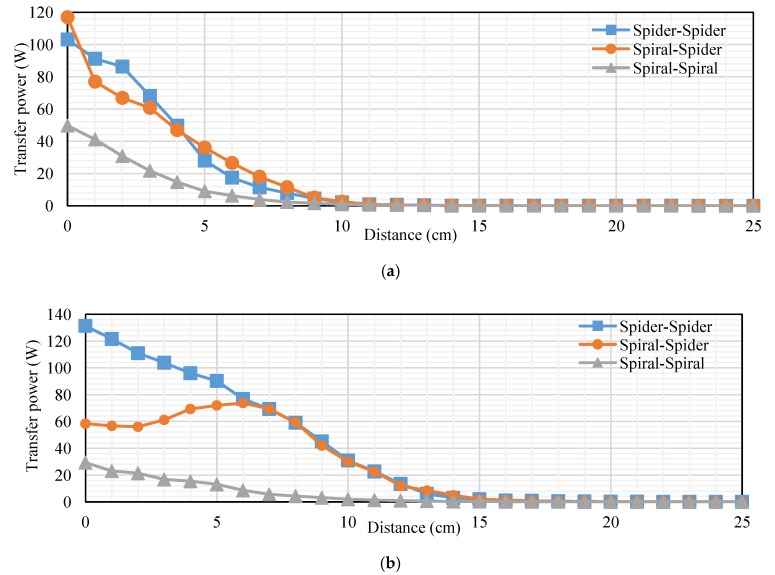
Transfer power on Rx_C_ at RL based on topologies at (**a**) 40 Ω and (**b**) 200 Ω.

**Table 1 sensors-20-02549-t001:** Specification for transmitter coils.

Parameters	Unit	Value for Transmitter Coil	
		Spiral	Spider
DC Input voltage	V	30	
AC out voltage	V	19	28
Operating frequency	kHz	13.6	13.6
American wire gage (AWG)	----	21	
Inductance	µH	695	720
Number of turns	turns	91	150
Diameter of turn	cm	18	19
Compensating capacitor	pF	150	

**Table 2 sensors-20-02549-t002:** Specification for receiver coils.

Parameters	Unit	Value for Receiver Coil	
		Spiral	Spider
AC out voltage	V	Based on distance	
Operating frequency	kHz	13.6	
American wire gage (AWG)	----	21	23
Inductance	µH	208	1000
Number of turns	turns	62	100
Diameter of turn	cm	12	15
Compensating capacitor	pF	150	

**Table 3 sensors-20-02549-t003:** The parameters for MRC based-loads at first topology.

	Resistive Load	@ 40 Ω		@ 50 Ω		@ 70 Ω	
Distances (cm)		Power (W)	Efficiency (%)	Power (W)	Efficiency (%)	Power (W)	Efficiency (%)
0		32.4	61.15	39.82	70.5	38.683	69.87
1		22.344	49.87	32.01	61.63	36.0144	68.21
2		14.234	43.16	21.087	50.11	24.8374	56.94
3		9.248	36.89	11.0026	37.63	13.025	40.66
4		5.356	29.6	6.885	31.11	8.24	34.48
5		4.169	27.1	4.173	26.26	5.136	28.01
6		1.7958	21.28	2.226	21.53	2.782	22.59
7		1.386	19.86	1.71	20.02	2.079	21.14
8		0.759	14.6	1.05	16.11	1.2533	13.01
9		0.5246	12.1	0.72	11.26	0.73112	7.59
10		0.476	6.28	0.5	6.53	0.4913	6.14
11		0.3	4.86	0.312	5.02	0.34435	3.7
12		0.195	2.32	0.2079	3.09	0.22794	2.36
13		0.1	1.55	0.156	2.31	0.16317	1.58
14		0.0756	0.82	0.0968	1.61	0.1148	1.11
15		0.0476	0.47	0.074	1.01	0.08365	0.73
16		0.0348	0.76	0.0496	0.67	0.0615	0.53
17		0.025	0.39	0.0378	0.5	0.04375	0.37
18		0.0189	0.3	0.0242	0.31	0.03066	0.27
19		0.01332	0.19	0.02	0.24	0.02413	0.2
20		0.00945	0.14	0.0144	0.16	0.0176	0.14
21		0.00702	0.1	0.0091	0.12	0.01209	0.1
22		0.00506	0.07	0.0072	0.09	0.00984	0.08
23		0.0036	0.05	0.00459	0.07	0.0072	0.06
24		0.0028	0.04	0.00344	0.05	0.00567	0.04
25		0.0021	0.03	0.0028	0.03	0.00378	0.03

**Table 4 sensors-20-02549-t004:** The parameters for MRC based-loads at second topology at RL.

	RL	25 Ω		33 Ω		50 Ω		100 Ω		200 Ω		300 Ω		400 Ω	
D (cm)		P (W)	ƞ (%)	P (W)	ƞ (%)	P(W)	ƞ (%)	P (W)	ƞ (%)	P (W)	ƞ (%)	P (W)	ƞ (%)	P (W)	ƞ (%)
0		10.56	89.88	14.1	96.27	10.85	73.48	13	97.96	17.5	98.16	14.85	97.12	16	97.5
1		7.5	83.96	10.906	90.48	9.06	61.12	10	97.93	12.6	96.44	13.44	96.76	13	97.4
2		3.653	73.9	8.494	88.52	7.1	54.84	9	96.55	11.4	96.37	11	95.83	12	94.7
3		2.08	62.94	5.5	74.54	5.2	40.33	7.5	95.56	9.26	95.55	8.4	95.51	10	93.1
4		1.68	44.64	3.52	59.14	4.08	38.18	6.2	91.21	7.9	93.95	7.426	95.45	8.7	93.1
5		0.95	31.79	2.136	36.51	3.04	33.11	4.8	76.45	6.75	93.37	6.6	94.5	7.2	92.1
6		0.616	17.95	1.5	24.89	1.963	23.76	3.6	57.74	5.74	92.41	5.332	94.48	5.5	89.2
7		0.3565	8.33	0.88	16.24	1.32	14.36	2.3	34.57	4.08	86.63	4.4	93.73	4	87.1
8		0.23	5.58	0.644	11.4	0.926	6.97	1.4	18.94	2.7	78.67	3.78	91.69	3.6	82.1
9		0.15	2.23	0.375	6.48	0.64	4.06	1	11.89	2.08	60.62	2.7	86.01	2.8	77.5
10		0.10168	0.7	0.248	3	0.4066	1.58	0.6	6.49	1.2	41.4	1.87	60.55	2	75.7
11		0.07038	0.37	0.17085	1.28	0.2759	0.93	0.4	4.19	0.8	31	1.107	40.01	1.2	60.9
12		0.04704	0.18	0.12056	0.9	0.2025	0.46	0.3	1.82	0.54	18.6	0.714	22.13	0.9	43.9
13		0.036	0.11	0.08388	0.51	0.152	0.27	0.2	1.19	0.37	8.27	0.504	14.58	0.5	22.8
14		0.0248	0.04	0.06	0.07	0.096	0.24	0.2	0.68	0.28	4.14	0.347	10.29	0.4	14.7
15		0.0182	0.04	0.04368	0.06	0.07	0.21	0.1	0.43	0.19	2.82	0.252	7.14	0.3	8.24
16		0.0132	0.03	0.03146	0.06	0.0516	0.09	0.1	0.26	0.15	1.46	0.189	3.25	0.2	5.39
17		0.00969	0.03	0.02337	0.05	0.042	0.08	0.1	0.23	0.11	1.08	0.139	1.82	0.2	3.41
18		0.00748	0.02	0.01785	0.04	0.0288	0.07	0.1	0.2	0.08	0.61	0.101	1.37	0.1	2.13
19		0.006	0.02	0.0138	0.04	0.0224	0.06	0.1	0.09	0.07	0.42	0.081	0.78	0.1	1.39
20		0.00442	0.02	0.0104	0.03	0.0168	0.05	0.1	0.07	0.05	0.25	0.06	0.54	0.1	0.82
21		0.0033	0.02	0.0077	0.03	0.012	0.05	0.1	0.06	0.04	0.23	0.051	0.33	0.1	0.72
22		0.0026	0.01	0.006	0.03	0.0086	0.04	0.1	0.06	0.03	0.1	0.04	0.28	0.1	0.49
23		0.00207	0.01	0.00486	0.02	0.008	0.04	0.1	0.05	0.02	0.08	0.028	0.24	0.1	0.29
24		0.00168	0.01	0.004	0.02	0.0068	0.03	0.1	0.05	0.02	0.08	0.022	0.23	0.1	0.26
25		0.00133	0.01	0.00294	0.02	0.0056	0.03	0.1	0.04	0.01	0.07	0.018	0.2	0.1	0.23

RL: Resistive Lode; D: Distance (cm); P: Power; ƞ: Efficiency.

**Table 5 sensors-20-02549-t005:** The parameters for MRC based-loads at third topology at RL.

	RL	25 Ω		33 Ω		50 Ω		100 Ω		200 Ω		300 Ω		400 Ω	
D (cm)		P (W)	ƞ (%)	P (W)	ƞ (%)	P (W)	ƞ (%)	P (W)	ƞ (%)	P (W)	ƞ (%)	P (W)	ƞ (%)	P (W)	ƞ (%)
0		30.24	98.26	54	97.5	59.16	97.63	19.78	99.84	19.56	98.98	6.768	98.63	5.445	98.11
1		22.814	92.44	40.4	96.25	46.41	94.92	19.125	99.03	17.05	96.27	6.3	97.71	5.04	96.6
2		15.5	85.03	28.8	95.71	38.07	93.32	18.06	97.6	13.5	93.33	5.2	96.43	4.18	93.58
3		10.44	82.55	24.64	93.54	29.6	92.2	17.22	86.94	9.03	92.73	5.88	95.07	4.1	92.08
4		6.656	53.26	13.65	80.48	23.45	86.21	16.8	83.57	9.68	90	6.45	95	4.73	90.57
5		5.4	40.33	10.08	65.45	12.5	83.04	15.8	76.05	9.9	87.27	6.975	94.93	4.84	89.23
6		4	29.41	5.6	51.15	7.372	57.27	10.4	67.6	9.03	84.25	7.2	91.55	4.95	87.69
7		2.48	17.54	3.52	35.29	4.47	40.36	6.25	66.87	7.03	82.5	6.776	86.63	4.95	86.15
8		1.5325	9.07	2.25	22.55	2.76	24.55	4	63.27	4.5	72.79	5.2	85.91	4.452	87.84
9		1.05	3.73	1.4	10	1.95	15.38	2.72	47.27	3.05	70	3.52	85.95	3.534	86.27
10		0.672	2.13	0.96	4.94	1.2	7.67	1.742	30.37	2	54.55	2.34	71.35	2.387	80
11		0.455	1	0.65	1.88	0.78	3.69	1.1	16.67	1.394	40.91	1.47	56.54	1.575	71.37
12		0.345	0.53	0.44	0.98	0.55	2.42	0.81	9.43	0.98	22.22	1.02	38.46	1	50.4
13		0.198	0.28	0.27	0.71	0.396	1.62	0.518	3.77	0.72	15.56	0.75	27.45	0.792	40
14		0.15	0.27	0.225	0.47	0.2775	0.77	0.378	1.42	0.5	9.26	0.48	17.65	0.54	27.2
15		0.0915	0.22	0.128	0.24	0.189	0.52	0.26	0.68	0.336	3.7	0.385	9.8	0.416	16.2
16		0.068	0.18	0.0972	0.2	0.1378	0.27	0.225	0.57	0.28	2.37	0.31	3.92	0.297	9.2
17		0.0495	0.18	0.0675	0.17	0.099	0.23	0.156	0.34	0.186	1.56	0.216	2.82	0.24	4
18		0.04	0.13	0.052	0.16	0.0741	0.19	0.096	0.26	0.13	0.81	0.161	1.76	0.168	2.88
19		0.0297	0.06	0.0408	0.13	0.0544	0.08	0.0784	0.25	0.1058	0.56	0.126	1.27	0.126	1.8
20		0.025	0.05	0.03	0.12	0.042	0.07	0.0625	0.11	0.08	0.33	0.09	0.94	0.096	1.3
21		0.021	0.05	0.0221	0.11	0.0312	0.07	0.0484	0.09	0.0648	0.3	0.0675	0.59	0.077	0.96
22		0.0132	0.04	0.01679	0.05	0.0253	0.06	0.0361	0.08	0.048	0.26	0.056	0.53	0.06	0.6
23		0.009	0.04	0.013	0.04	0.02	0.05	0.0272	0.08	0.0392	0.23	0.042	0.31	0.0495	0.54
24		0.007	0.04	0.01026	0.04	0.0162	0.05	0.0225	0.07	0.0288	0.21	0.0352	0.27	0.04	0.32
25		0.005	0.03	0.00816	0.04	0.0112	0.04	0.0182	0.06	0.0242	0.19	0.028	0.24	0.0315	0.28

RL: Resistive Lode; D: Distance (cm); P: Power; ƞ: Efficiency.

**Table 6 sensors-20-02549-t006:** Performance metrics from past studies.

Ref/Year	Objective	Operating Frequency (MHz)	Implementation Environment	Application	Transfer Distance (cm)	Transfer Efficiency (%)	Transfer Power (W)
[12]/2009	OTE	7	Experimental	Id	9	22.3	N/A
[8]/2010	OTE	0.7	Experimental	Id	0.32	72	N/A
[11]/2011	OTE	0.00988	Simulation by using Proteus	LVAD	N/A	N/A	1.5
[7]/2013	OTE	0.9532	Experimental	light neon, and lamp	1–15	N/A	N/A
[9]/2014	OTE	2.75	Experimental	Prototype	3	93.1	N/A
[10]/2014	OTE	16.47	Experimental	capsule endoscopy	7	0.71	N/A
[14]/2014	OTDE	0.3	Experimental	implantable cardioverter defibrillators	6	7	1
[13]/2015	OTDE	403	Experimental	pacemaker	10	5.24	10 × 10^−6^ 1 × 10^−3^
[15]/2016	battery-recharge	0.3–13.56	Experimental	pacemaker	10	N/A	0.0004
[17]/2016	OTDE	0.02	Simulation by multiphysics COMSOL	pacemaker	41	N/A	N/A
[18]/2016	OTE	137	Experimental	Brain region	2.8	0.76 @air 0.6@ lamb head	240 × 10^−6^ 191 × 10^−6^
[19]/2018	OTDE	1	Experimental	home appliance operations	2 6	80.66 66.66	4.84 4
Spiral–spiral topology	OTDE	0.014	Experimental	Heart rate sensor	5 8 10	28.8 13 6.4	5 1.3 0.5
Spider- spidertopology	OTDE	0.014	Experimental	Heart rate sensor	5 8 10	92.11 82.02 75.8	7.2 3.6 2.04
Spiral–spider topology	OTDE	0.014	Experimental	Heart rate sensor	5 8 10	90 88 80	5 4.5 2.5

N/A: not available; OTE: optimization transfer efficiency; OTDE: optimize transfer distance and efficiency.

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
