# Peer review of "Hybrid Coils-Based Wireless Power Transfer for Intelligent Sensors"

_sensors, 2020, doi:10.3390/s20092549_

Round 1

Reviewer 1 Report

In this paper authors proposed an energy harvesting technique based on wireless power transfer/magnetic resonator coupling for biomedical devices. They studied the power transfer and efficiency of biomedical transmitter and receiver coils at different distances.

The introduction must be rewritten to clarify what are the contributions of this work related with others work that are cited. Moreover, this paper is a practical experience report where authors report many results from an experimental setup. It just reports the design of electromagnetic coils and test it.

The discussion of subsection 6.1.3. must be rewritten to simplify its reading. I suggest using a Table to represent the values.
Figure 12 is not easy to understand. Is the color scale the same for the three situations?

What are the reasons that originates high discrepancies between low RL and high RL? Which type of load is used (resistive or inductive)? What is the difference between them in the efficiency?

The results reported in Figure 16 and its discussion should be presented in other way (perhaps a table).

Nome of the 3D charts include a color grade. Is the same grade used for all charts?

In figure 22, an explanation about the performance of Spiral-Spiral approach is missing. Why this low performance is achieved?

What is the motivation of the behavior of Spiral-Spider curve in figure 23 b). Why is it different from other approaches?

Author Response

Dear Reviewer 1,

We greatly appreciate your thoughtful comments and suggestions that helped improve our manuscript. We have addressed all the comments as explained below and believed that the revised manuscript can meet the journal publication requirements.

Best regards

Authors

Comments and Suggestions for Authors

In this paper authors proposed an energy harvesting technique based on wireless power transfer/magnetic resonator coupling for biomedical devices. They studied the power transfer and efficiency of biomedical transmitter and receiver coils at different distances.

# The introduction must be rewritten to clarify what are the contributions of this work related with others work that are cited. Moreover, this paper is a practical experience report where authors report many results from an experimental setup. It just reports the design of electromagnetic coils and test it.

Thank you very much for your valuable comment. According to the reviewer's suggestion, we have added a paragraph in the introduction (please see section 1, page 2, lines 79-88). Also, it can be found in section 8, page 8, lines 317-322.

# The discussion of subsection 6.1.3. must be rewritten to simplify its reading. I suggest using a Table to represent the values. Figure 12 is not easy to understand. Is the color scale the same for the three situations?

We have followed this recommendation, this section has been rewritten and added a table of values, In addition, it has been highlighted in subsection 6.1.3, it can be found in page 12, lines 422-423.

# What are the reasons that originates high discrepancies between low RL and high RL? Which type of load is used (resistive or inductive)? What is the difference between them in the efficiency?

According to the reviewer's suggestion , the contents of the paper have been added to the paper. It can be found in section highlighted in subsection 6.2.2 page 14, lines 467-470.

# The results reported in Figure 16 and its discussion should be presented in other way (perhaps a table).

We have followed this recommendation, this section has been rewritten and added a table of values.  In addition it has been highlighted in subsection 6.2.3, it can be found in page 15-16, lines 489-490. Moreover, it has been highlighted in subsection 6.3.3, it can be found in page 18-19, lines 555-556.

# Nome of the 3D charts include a color grade. Is the same grade used for all charts?

Done, this section has been rewritten and added table of values, In addition it has been highlighted in subsection 6.1.3, it can be found in page 12, lines 422-423. Moreover, it has been highlighted in subsection 6.2.3, it can be found in page 15-16, lines 489-490.

# In figure 22, an explanation about the performance of Spiral-Spiral approach is missing. Why this low performance is achieved?

We have followed this recommendation (please see Section 7, page 20, lines 5585-587).

#What is the motivation of the behavior of Spiral-Spider curve in figure 23 b). Why is it different from other approaches?

We have followed this recommendation (please see Section 7, page 21, lines 601-604).

Many thanks for your time and effort,

Reviewer 2 Report

This paper presents a hybrid coils-based wireless power transfer for biomedical sensors. Two concerns are provided below.

(1) The motivation of this work is not convincing. The authors claim that this system is for biomedical sensors. However, both transmitting and receiving coils are in free space, and there is no biomedical tissue between them. Also, the coil sizes seem a little large. In biomedical applications, one coil could be implanted, so the size should be miniaturized. Therefore, the wireless power transfer system cannot be used for biomedical applications.

(2) The authors report the measured results of the wireless power transfer system. Why not simulate the transmitting and receiving coils and optimize the power transfer performance? So that the optimized can be obtained. In this work, only several discrete parameters of the system are measured and the optimized performance could be missed.

Author Response

First, the authors would like to thank the reviewers and editor for their valuable comments and contributions to our manuscript. We find that the comments greatly improve the accuracy and clarity of the paper. We have revised the manuscript in accordance to the reviewer’s comments. Please find the author’s response to each of the reviewer comments below:

Reviewer 2

Comments and Suggestions for Authors

Comment 1: The motivation of this work is not convincing. The authors claim that this system is for biomedical sensors. However, both transmitting and receiving coils are in free space, and there is no biomedical tissue between them. Also, the coil sizes seem a little large. In biomedical applications, one coil could be implanted, so the size should be miniaturized. Therefore, the wireless power transfer system cannot be used for biomedical applications.

Comment 1 A: The motivation of this work is not convincing

Response 1 A:

Thank you very much for your valuable comment. According to the reviewer's suggestion, we have added a paragraph in the introduction (please see section 1, page 2, lines 79-88). Also, it can be found in section 8, page 8, lines 317-322.

Comment 1 B: The authors claim that this system is for biomedical sensors.

 Response 1 B: The highlight of the paper has been added to the paper. It can be found in section 5, page 8, lines 312-317. In addition, It can be found in section 5, page 9, lines 334-337. Moreover, It can be found in section 5, page 10, lines 342-347. Furthermore, it has been highlighted in subsection 6.1.2, it can be found in page 11, lines 373-382. Further, it has been highlighted in subsection 6.2, it can be found in pages 13-14, lines 434-444. Also, it has been highlighted in subsection 6.3.1, it can be found in page 16, lines 501-511.

Comment 1 C: However, both transmitting and receiving coils are in free space, and there is no biomedical tissue between them.

Response 1 C: The contents of the paper have been added to the paper. It can be found in Section 9, page 23, lines 643-646.

Comment 1 D: Also, the coil sizes seem a little large.

Response 1 D: We have followed this recommendation (please see section 4, pages 5-6, lines 225-239).

Comment 2: The authors report the measured results of the wireless power transfer system. Why not simulate the transmitting and receiving coils and optimize the power transfer performance? So that the optimized can be obtained. In this work, only several discrete parameters of the system are measured and the optimized performance could be missed.

Response 2: We have followed this recommendation (please see section 9, page 23, lines 646-647.

Many thanks for your time and effort,

Round 2

Reviewer 1 Report

The introduction is not clear. The listed contributions (bullets) must be explained in deep, referring why previous works do not solve it. 

Author Response

Dear Reviewer 1,

We greatly appreciate your thoughtful comments and suggestions that helped improve our manuscript. We have addressed all the comments as explained below and believed that the revised manuscript can meet the journal publication requirements.

Best regards

Authors

Comments and Suggestions for Authors

#1 A: The introduction is not clear.

Thank you very much for your valuable comment. According to the reviewer's suggestion, we have modified a paragraph in the introduction (please see section 1).

#1 B: The listed contributions (bullets) must be explained in deep, referring why previous works do not solve it. 

According to the reviewer's suggestion , we have modified to the paper. It can be found in section the introduction (please see section 1, lines 76-86). Also, it can be found in section Comparison with Previous Work (please see Table 6).

Many thanks for your time and effort,

Reviewer 2 Report

Thanks to the authors for the efforts in improving the manuscript. However, I still think the review's concerns have not been addressed. For some problems, the authors did not modify the work and manuscript, or just write that will be explored in future work.

The unsolved problems are given as follow.

  1. The authors claim that this system is for biomedical sensors. However, both transmitting and receiving coils are in free space, and there is no biomedical tissue between them.
  2. The coil sizes seem a little large. In biomedical applications, one coil could be implanted, so the size should be miniaturized. Therefore, the wireless power transfer system cannot be used for biomedical applications.

Author Response

Dear Reviewer 2,

We greatly appreciate your thoughtful comments and suggestions that helped improve our manuscript. We have addressed all the comments as explained below and believed that the revised manuscript can meet the journal publication requirements.

Best regards

Authors

Comments and Suggestions for Authors

#1: The authors claim that this system is for biomedical sensors. However, both transmitting and receiving coils are in free space, and there is no biomedical tissue between them.

Thank you very much for your valuable comment. According to this comment, we have changed the title of our manuscript to “Hybrid Coils-based Wireless power transfer for Intelligent Sensors”.

#2: The coil sizes seem a little large. In biomedical applications, one coil could be implanted, so the size should be miniaturized. Therefore, the wireless power transfer system cannot be used for biomedical applications.

Thank you for your suggestion, For the received coils, these coils are a prototype for which the number and the distance between the turns can be regulated by using a specific machine or using printed coils in future applications to reduce the size and weight for the purpose of implanting inside the body. Future work will focus on design and implement a system with an integrated antenna in a printed spiral and spider coils to reduce the size and weight of the implantable device. However, we have changed the title of our manuscript to “Hybrid Coils-based Wireless power transfer for Intelligent Sensors”

Many thanks for your time and effort,

Round 3

Reviewer 1 Report

Thanks for the revised version. Now, the paper is more clear.

Reviewer 2 Report

Thanks to the authors for the revisions.